# Ammonia dimer: extremely fluxional but still hydrogen bonded

Aling Jing[1✉], Krzysztof Szalewicz [1✉] & Ad van der Avoird [2✉]

In the 1980s, Nelson, Fraser, and Klemperer (NFK) published an experimentally derived structure of the ammonia dimer dramatically different from the structure determined computationally, which led these authors to the question "Does ammonia hydrogen bond?". This question has not yet been answered satisfactorily. To answer it, we have developed an ab initio potential energy surface (PES) for this dimer at the limits of the current computational capabilities and performed essentially exact six-dimensional calculations of the vibration-rotation-tunneling (VRT) spectra of $NH_3$-$NH_3$ and $ND_3$-$ND_3$, obtaining an unprecedented agreement with experimental spectra. In agreement with other recent electronic structure calculations, the global minimum on the PES is in a substantially bent hydrogen-bonded configuration. Since the bottom of the PES is exceptionally flat, the dimer is extremely fluxional and the probability of finding it in configurations that are not hydrogen bonded is high. Nevertheless, the probability of hydrogen-bonded configurations is large enough to consider the ammonia dimer to be hydrogen bonded. We also show that NFK's inference that the ammonia dimer is nearly rigid actually results from unusual cancellations between quantum effects that generate differences in spectra of different isotopologues.

[1] Department of Physics and Astronomy, University of Delaware, Newark, DE 19716, USA. [2] Theoretical Chemistry, Institute for Molecules and Materials, Radboud University, Heyendaalseweg 135, 6525 AJ Nijmegen, The Netherlands. ✉email: allinson@udel.edu; szalewic@udel.edu; a.vanderavoird@theochem.ru.nl

As computational predictions of properties and design of materials enter the mainstream of research[1–4], it is important to fully understand the underlying physics. In particular, structural predictions are in most cases governed by noncovalent interactions. Therefore, it is essential that theory and experiment are fully reconciled on small clusters[5], a class of systems where the two approaches can achieve a very high accuracy of computed and measured properties, respectively. One example of an outstanding disagreement (35 year old) is the ammonia dimer, for which there are still several unanswered questions. In particular, there is disagreement concerning the structure of the ammonia dimer, clearly seen in Fig. 1. The most striking difference is that the theoretical structure[6] is a linear-staggered one, i.e., is hydrogen bonded, while the experimental one[7–9] is not. The experimental evidence made NFK to conclude[9] that "NH₃ might well be best described as a powerful hydrogen-bond acceptor with little propensity to donate hydrogen bonds". Also, the measured dipole moment of the dimer, 0.74 D, was much smaller than the dipole moment of about 2 D of the theoretical equilibrium structure. In the early 1990s, Havenith et al.[10] and Loeser et al.[11] extended the experimental information on the ammonia dimer by measuring its VRT spectra; however, this work has not resolved the discrepancies. Most recent experimental work on the ammonia dimer involved recording spectra in helium nanodroplets[12,13] and measuring the dissociation energy[14]. In view of the ubiquity of ammonia molecules both terrestrially and extraterrestrially[15], as well as their significance in industrial and pharmaceutical synthesis, in living organisms, and as a "green" fuel[16], understanding of the interactions between such molecules is critically important.

For floppy dimers, the structure deduced from experiments does not need to be close to the equilibrium structure on the potential energy surface (PES). However, the NFK experiment[7–9] provided arguments that the ammonia dimer is quite rigid. This conclusion was questioned by one of the present authors and coworkers[17–19] based on calculations of dimer VRT spectra. These authors have also shown that ammonia-dimer PESs with hydrogen-bonded equilibrium structures can have vibrationally averaged structures that are not hydrogen bonded. However, the first-principles PESs available at that time were of insufficient quality and most findings were based on an empirical surface fitted to experimental spectra, which might lead to biased conclusions. The 1980s computational results concerning the dimer structure were mostly confirmed in the early 2000s by computations[20] that used higher levels of electronic structure theory, larger basis sets, and more advanced methods of searching for stationary points (SPs). The only essential difference was that the equilibrium hydrogen bond is substantially bent. However, the revised theoretical structure is still is very far from the NFK structure. Whereas these computations established beyond doubt the position of the minimum on the PES, they could not be considered as a proof that the ammonia dimer is hydrogen bonded since the probability of finding the dimer in this geometry could be small.

## Results and discussion

**Physical origins of unusual features on PES.** The best published first-principles PES for the ammonia dimer originates from the 1980s[21] and has uncertainties of the order of 1 kcal/mol. Our goal

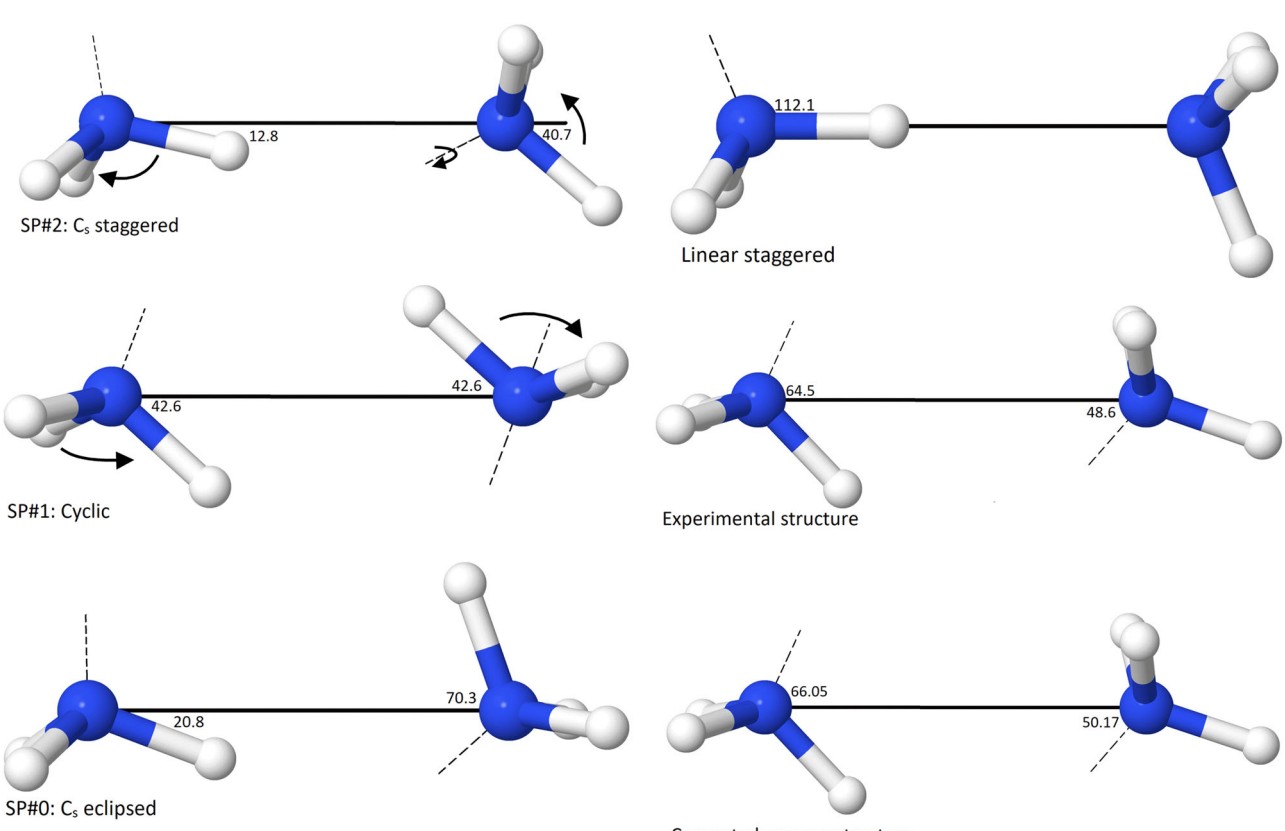

**Fig. 1 Structures of the ammonia dimer.** Left column: stationary points. Right column: the linear hydrogen-bonded structure predicted by the 1980s theory[6], the experimental structure[7–9], and the average structure from the present work. In the left column, the numbers are ∠HNN angles in degrees, except for SP#2 monomer B, where 180-∠HNN is listed. In the right column, the angles shown for the experimental and the average structures are between the C₃ axes of monomers and the principal rotational axis of the complex. For the linear structure, the C₃-NH angle is shown. The arrows visualize the motions on the paths from SP#1 and SP#2 to SP#0.

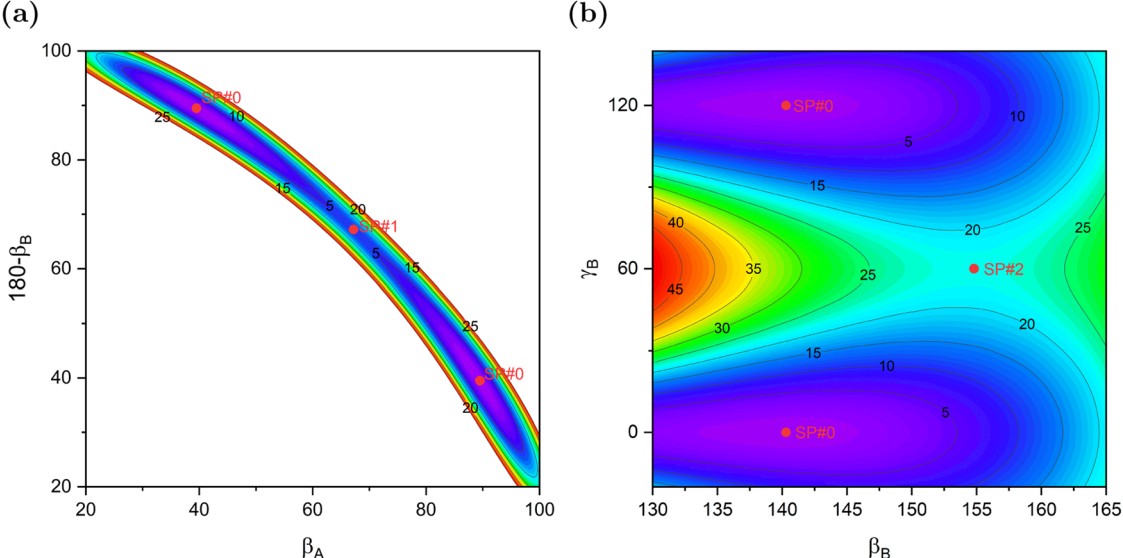

**Fig. 2 Shape of the potential energy surface.** Contour plots [cm⁻¹] of the PES in the region of the global minimum, SP#0, and the lowest saddle points, SP#1 and SP#2, with the interaction energy shifted to be zero at the global minimum. Panel **a** displays the region of the hydrogen-bond donor–acceptor interchange path between two symmetry-equivalent minima, SP#0, through SP#1. The Euler angles $\beta_A$ and $\beta_B$ are the angles (in degrees) between the $C_3$ axes of the monomers and the vector **R** joining the COMs of the monomers and oriented from A to B. The remaining Euler angles $\alpha_B = 180°$, $\gamma_A = 60°$, and $\gamma_B = 0°$ are fixed at their equilibrium values and the distance $R$ is optimized to minimize the interaction energy. Panel **b** displays the $\gamma_B$ rotation of the acceptor monomer about its $C_3$ axis through the saddle point SP#2 for a range of $\beta_B$ values. The angles $\alpha_B = 180°$ and $\gamma_A = 60°$ are fixed, $\beta_A$ and $R$ are optimized.

was to develop a surface with uncertainties 100 times smaller and use it in quantum nuclear dynamics calculations of the dimer spectra. The ab initio coupled cluster calculations with single, double, and noniterative triple excitations, CCSD(T), and the fit procedure by which we achieved this goal are described in Methods. Additional calculations at higher levels of theory established the uncertainty of our PES relative to the exact interaction energy to be about 0.02 kcal/mol.

The dimer geometries can be described in terms of the distance between the centers of mass (COMs) of the monomers, $R$, and five Euler angles, using the same convention as in ref. [22]. Figure 2 displays 2D cuts of the PES, with the remaining parameters fixed or optimized. Panel a shows that the ammonia-dimer PES is very unusual in that it has a very narrow canyon-like valley where the two minima are located. The bottom of the valley is almost flat since the SP#1 barrier is only 5.2 cm⁻¹. The energetics and geometries of SPs are given in Supplementary Table 1. Thus, the interconversion between the two minima along the bottom of the valley is very easy. Note that the molecules must rotate in a geared fashion since the valley is very narrow. Panel b shows that also the SP#2 barrier for rotation of the acceptor monomer about its $C_3$ axis is very low, only 22 cm⁻¹, which should be compared with the potential well depth $D_e$ of 1136 cm⁻¹. Thus, the region around the minima of the PES is very flat, which has to result in extraordinary floppiness of the dimer. The dimers at the discussed SP geometries and the experimental structure are depicted in Fig. 1. Supplementary Fig. 1 illustrates the motions on the path from SP#2 to SP#0. The equilibrium geometry at SP#0 can be considered a hydrogen-bonded configuration, although with a significantly bent bond, by 21°. This structure agrees reasonably well with the latest published ab initio calculations[20]. The comparison of the experimental structure with the structures at different stationary points shows that it is closest to the SP#2 structure.

To understand the origin of the narrow valley, we performed calculations in this region using symmetry-adapted perturbation

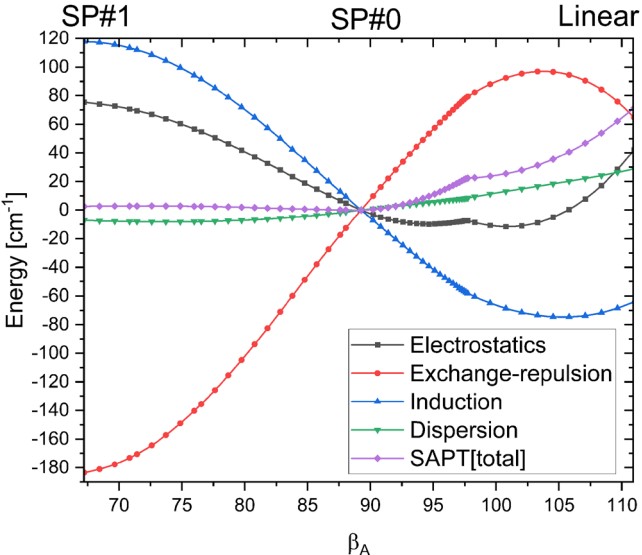

**Fig. 3 Physical components of interaction energy.** SAPT components, relative to their values at SP#0, along the path from SP#1 through SP#0 to the staggered linear hydrogen-bonded structure (see the discussion on the SAPT components in the Supplementary Information for more details).

theory (SAPT)[23], see Methods for details of these calculations. SAPT constructs the interaction energy as a sum of components with well-defined physical origins: electrostatic, exchange-repulsion, induction, and dispersion interactions. The behavior of these components along the path from SP#1 through SP#0 to the staggered linear hydrogen-bonded structure is shown in Fig. 3. As can be seen, the dispersion curve is essentially flat along the whole path, which is expected as the dispersion interaction is the most isotropic component. The electrostatic and induction components sum up to substantially favor SP#0 over SP#1, by 193 cm⁻¹.

The exchange-repulsion component, on the other hand, favors SP#1 over SP#0, and the competition between these effects leads to the very small barrier of only 3 cm$^{-1}$ at the SAPT level. The exchange-repulsion is roughly proportional to the overlap of the monomers' densities, which is smallest for the cyclic structure at SP#1. An overlap-driven variation of the components in the direction perpendicular to the path through SP#1, i.e., across the valley in Fig. 2a, explains the narrowness of the valley (see Supplementary Fig. 3 and its discussion). For the linear structure shown on the right-hand side of Fig. 3, the interplay of physical effects is almost the opposite of that discussed above for SP#1. The linear structure is disfavored relative to SP#0 by exchange since at the linear configuration the density overlap is at its maximum: the bonding hydrogen points directly into the high density region of the acceptor's lone pair. Since the changes of the three attractive components nearly cancel each other, the overall destabilization of the linear configuration is mostly due to the exchange interactions. The overall effect is that the linear structure lies 75 cm$^{-1}$ above the minimum (at the SAPT level). The balancing act between the four physical components of the interaction energy that takes place in the vicinity of the global minimum leads to an exceptionally flat PES in this region, explaining the difficulties encountered by theory in searching for the minimum geometry[6,20,21,24]. One may point out here that this analysis shows that also the formation of a hydrogen bond is due to this balancing act, and not to a single physical component. In particular, electrostatic interactions do not play a dominant role, in fact, electrostatics favors SP#2 over SP#0, see the Supplementary Information for a further discussion.

**Dimer spectra and properties**. The PES developed here was used in six-dimensional quantum calculations to generate the VRT spectrum of the ammonia dimer. Details of VRT calculations are given in Methods. The calculated and experimental energy levels for angular momenta $J = 0$, 1, and 2, labelled by the irreducible representations of the group $G_{36}$[18,25], are shown in Fig. 4. Although the Coriolis coupling between basis functions with different $K$ is exactly included, we also list the approximate $K$ quantum numbers derived from the analysis of the calculated wave functions. All levels that can be directly compared with experimental levels extracted from high-resolution spectra[11] are shown. The root-mean-square deviations (RMSDs) of theory with respect to experiment are 1.25, 0.94, and 1.08 cm$^{-1}$ for the states with $J = 0$, 1, and 2, respectively. Considering that the potential surface used to calculate the energy levels is completely ab initio, with no parameters fitted to experimental data as in ref. [18], the agreement of the calculated levels with the experimental data is good. To further improve the agreement with experiment, one would have to include the intramonomer degrees of freedom. However, while it would be possible to develop an 18-dimensional PES for the ammonia dimer (full-dimensional PESs have been developed for the water dimer and trimer[26–33]), accurate VRT calculations have been performed so far only for up to 12 degrees of freedom. Comparisons between 6- and 12-dimensional calculations for the water dimer[34] made in ref. [35] show only small monomer-flexibility effects on the transitions involving intermonomer motions that we consider here.

Also various properties of NH$_3$-NH$_3$ have been measured[7,8,11,36]. Table 1 compares these data with the quantities calculated with the potential. The calculated dipole moment is the length of the orientation-dependent dipole vector of the dimer averaged over the dimer wave functions. The dimer dipole vector is the vector sum of the permanent dipole moments of the NH$_3$ monomers plus the dipole-induced dipole moments on both monomers. The agreement of the calculated rotational constants $(B + C)/2$ with the

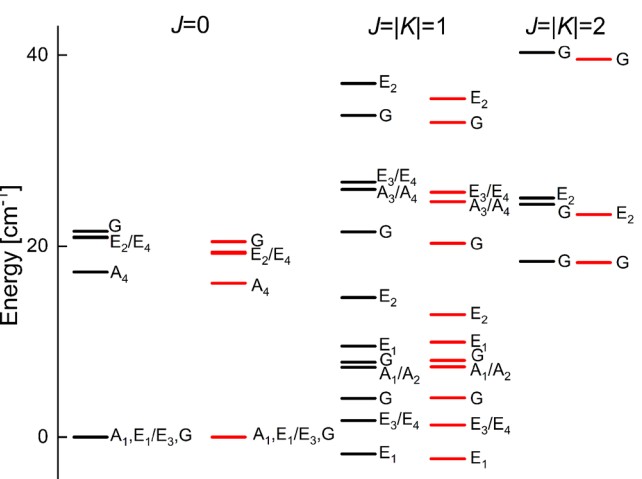

**Fig. 4 VRT energy levels of the ammonia dimer.** Calculated energy levels (black, in cm$^{-1}$) of NH$_3$-NH$_3$. In the rigid-monomer approach used, the permutation-inversion symmetry group is $G_{36}$. This group has four one-dimensional irreducible representations (irreps) $A_i$ with $i = 1, ..., 4$, four two-dimensional irreps $E_i$, $i = 1, ..., 4$, and one four-dimensional irrep $G$[18,25]. The irreps $A_i$ correspond to the monomer combination orthoNH$_3$-orthoNH$_3$, irreps $E_i$ to the paraNH$_3$-paraNH$_3$ combination, and irrep $G$ to the orthoNH$_3$-paraNH$_3$ combination, and vice-versa. The energy levels, given relative to the $J = 0$ ground-state levels of $A_1$, $E_1$, and $G$ symmetry, are compared with the experimental levels (red) from high-resolution spectra[10,11]. The numerical values are given in Supplementary Table 2.

**Table 1 Comparison of calculated properties of NH$_3$-NH$_3$ and ND$_3$-ND$_3$ with the available experimental data[7,8,11,36].**

| Symmetry | NH$_3$-NH$_3$ | | ND$_3$-ND$_3$ | |
| --- | --- | --- | --- | --- |
| | Theory | Experiment | Theory | Experiment |
| | Rotational constants $(B + C)/2$ of $K = 0$ states (MHz) | | | |
| $A_1$ | 5139.4 | 5136.6 | | |
| $A_4$ | 5025.0 | 5030.6 | | |
| $E_{1,3}$ | 5147.6 | 5115.6 | | |
| $E_{2,4}$ | 5048.0 | 5050.5 | | |
| $G$ | 5114.8 | 5110.5 | 4192.4 | 4190.3 |
| $G$ | 5163.6 | 5035.0 | | |
| | Dipole moment (Debye) | | | |
| | $K = 0$ states | | | |
| $G$ | 0.70 | 0.74 | 0.52 | 0.57 |
| | $|K| = 1$ states | | | |
| $G$ | 0.16 | 0.10 | | |
| $G$ | 0.26 | <0.09 | | |
| | Angles in lowest $G$ state with $J = K = 0$ | | | |
| $\theta_A$ | 66.05° | 64.5° | 63.95° | 62.6° |
| $180° - \theta_B$ | 50.17° | 48.6° | 51.55° | 49.6° |

Rotational constants have been extracted from the high-resolution spectra[11], the dipole moment has been obtained from Stark measurements[7,8,36], and the average angles $\theta_A$ and $\theta_B$ have been obtained from $^{14}$N nuclear quadrupole splittings observed in the microwave spectra[7,8], where $\theta_X$ is the angle between the $C_3$ axis of monomer $X = A$ or $B$ and the principal $a$ axis of the complex. This principal axis nearly coincides with the intermolecular axis **R** and the angles $\theta_X$ are practically the same as the Euler angles $\beta_X$. The $E_1$, $E_3$ states and the $E_2$, $E_4$ states are combined because only a single experimental $(B + C)/2$ value is available for these states.

experimental values is excellent: the differences are only about 0.1%. Exceptions are the values for the $E_{1,3}$ states and the second $G$ state. These exceptions can be explained because these particular $K = 0$ states are strongly perturbed by Coriolis coupling with $|K| = 1$ states of the same symmetry, which have nearly the same energy. The overall good agreement for the rotational constants shows that our potential produces realistic values of the averaged

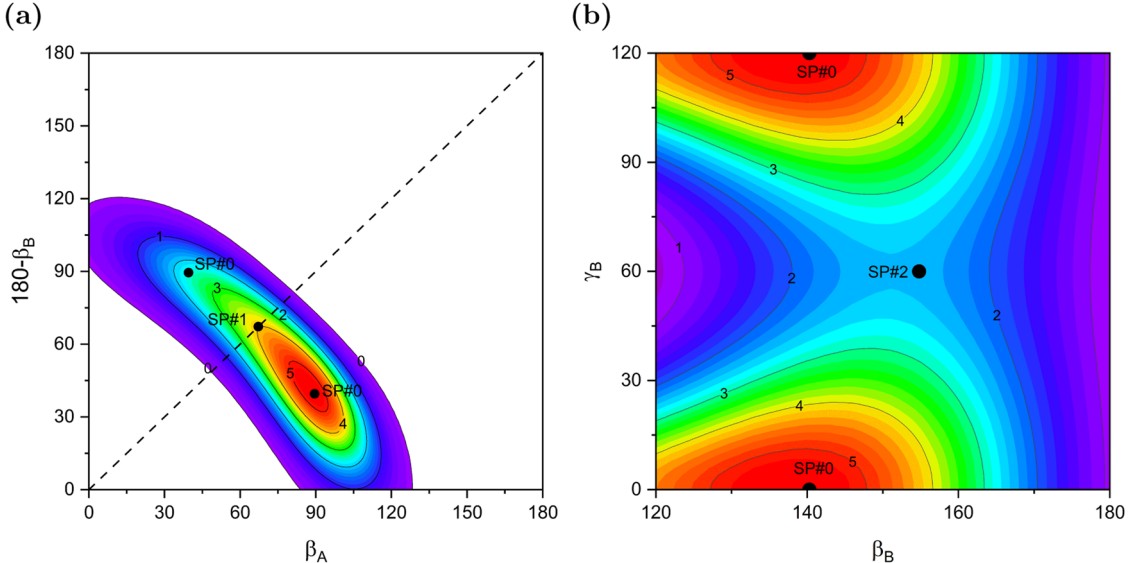

**Fig. 5 NH₃–NH₃ ground-state wave function of G symmetry.** The absolute value squared is plotted in bohr$^{-3}$. Panel **a** shows the region of the interchange path through SP#1 for which the PES is shown in Fig. 2a. Panel **b** shows the SP#2 region for which the PES is shown in Fig. 2b. Note that the optimized values of $\beta_A$ are chosen to be around 90°, so that the wave function in panel **b** contains (twice) the highest maximum shown in panel **a** with $180° - \beta_B \approx 40°$.

intermolecular bond lengths, $R_0$, for various states of different symmetry, which confirms that also the computed equilibrium bond length $R_e$ corresponding to the global minima in the potential and the $R$ values corresponding to the lower saddle points (where the wave function still has significant amplitude) are very accurate. The calculated dipole moments of different dimer states with $K = 0$ and $|K| = 1$ agree fairly well with the experimental data. The average angles $\theta_A$ and $\theta_B$ that were measured for the ground state of $G$ symmetry are very well reproduced by our calculated values, which confirms that also the anisotropy of the ab initio potential was accurately computed.

Another property of NH₃–NH₃ that has been measured[14] is the dissociation energy: $D_0 = 660 \pm 20$ cm$^{-1}$. Actually, the value of $D_0$ is different for the different nuclear spin species. The $D_0$ values are calculated from the ground-state energies $E_0$ taking into account that the nuclear spin multiplets are conserved upon dissociation and that with the neglect of umbrella inversion tunneling *ortho*NH₃ has a ground-state $j_k = 0_0$ with rotational energy 0, while *para*NH₃ has a ground-state $j_k = 1_1$ with rotational energy 16.2446 cm$^{-1}$. The $E_0$ and $D_0$ values for the different nuclear spin species are given in Supplementary Table 2. It is not clear from ref. [14] to which symmetry species the measured value refers, but from the procedure described in this paper it might be deduced that it is probably the lowest value that corresponds to the most abundant $A_1$ (*ortho-ortho*) species. With this assumption, we observe that the calculated $D_0$ value is 4.7% larger than the experimental one and is 1.6% above the experimental uncertainty. This is about the same relative discrepancy as for VRT energy levels. As in the latter case, the $D_0$ discrepancy is probably due to the assumption of monomer rigidity. Although the full-dimensionality treatment tends to increase $D_0$ for the water dimer[35], this effect may be of the opposite sign for the ammonia dimer due to the umbrella motions in the monomers.

We have also performed calculations for ND₃–ND₃, which are critical for reinterpretation of the spectroscopic experiments. The high-resolution spectrum of ND₃–ND₃ has not yet been measured, but some of its properties have been determined experimentally[8] and are listed in Table 1. Also here we find excellent agreement between the calculated rotational constant and the measured value, as well as between the calculated angles $\theta_A$ and $\theta_B$ and the values obtained from nuclear quadrupole

splittings. This shows that the wave functions obtained from the ab initio potentials accurately reproduce the geometrical distribution of the monomers in the complex, both for NH₃–NH₃ and ND₃–ND₃. As for NH₃–NH₃, the calculated dipole moment agrees quite well with the experimental value.

**Quantum origins of apparent dimer rigidity.** To understand the isotope effect in the dipole moment and in the angles $\theta_A$ and $180° - \theta_B$, one must realize that the potential has 18 global minima that are pairwise related through the interchange of the donor and acceptor in the hydrogen bond. Two related minima each correspond to a strongly bent hydrogen bond and the donor–acceptor interchange path between them passes over a saddle point of type SP#1. The potential itself is symmetric with respect to this interchange, but the dimer states of $G$ symmetry studied by NFK[7,8] are slightly asymmetric because the NH₃ (or ND₃) monomers in these states belong to different species: *ortho* and *para*, which have different rotational wave functions. This is why the dimer $G$ states have a measurable dipole moment and different average angles $\theta_A$ and $180° - \theta_B$. The ground-state wave function of $G$ symmetry, although delocalized over the two equivalent minima, is asymmetric and favors one minimum over the other. This is illustrated in Fig. 5a. For ND₃–ND₃, because of its smaller monomer rotational constants, the ground-state wave function is more strongly localized and has a larger amplitude at the favored minimum, see Supplementary Fig. 4(a). Due to this effect, one would expect the dipole moment and the difference between the angles $\theta_A$ and $180° - \theta_B$ to be larger than for NH₃–NH₃. Actually the dipole moment is smaller for ND₃–ND₃ than for NH₃–NH₃, and also the difference between $\theta_A$ and $180° - \theta_B$ is smaller. This is because of a second effect which is also related to the smaller rotational constants of ND₃. The different dynamical behavior of *ortho* and *para* monomers in the same potential is a pure quantum effect. It is smaller for ND₃ than for NH₃, because of its smaller rotational constants and larger moments of inertia, which make it behave more classically. Because of this smaller *ortho-para* difference, the $G$ symmetry wave functions of ND₃–ND₃ are more localized towards the symmetric SP#1 structure than those of NH₃–NH₃. This second

effect works in the direction opposite to the first one: it makes the $G$ states less asymmetric and thus reduces the dipole moment and the difference between $\theta_A$ and $180° - \theta_B$. Our calculations show, in agreement with experiment, that the second effect dominates. The small shifts in the angles from $NH_3$–$NH_3$ to $ND_3$–$ND_3$ that led NFK[7,8] to the conclusion that the ammonia dimer is quite rigid are actually caused by two quantum dynamical effects that work in opposite directions.

**Importance of SP#2.** As already mentioned, the experimental structure is closest to the structure of SP#2, see Fig. 1. Since the SP#2 barrier is only 22 cm⁻¹, it lies well below the ground rovibrational state level and therefore a significant fraction of the ground-state probability is located in this region, as seen in Fig. 5b. Comparisons of panels a and b shows that the magnitude of the wave function is larger at SP#1 (around 4 bohr⁻³) than at SP#2 (around 2.5 bohr⁻³), but the integration of this quantity will lead to a larger probability in the region around the SP in the latter case due to the shape of the wave function. Thus, although the only interconversion path discussed in the literature was the path through SP#1, it appears that the path through SP#2 is also significant.

**Concluding remarks.** Our work resolves long-lasting discrepancies between theory and experiment concerning the ammonia dimer. We found that the PES of this system, computed with uncertainties about two orders of magnitude smaller than those of the best published PES, is very flat at the bottom for motions in some coordinates, but very steep in some other ones, with strong couplings between these coordinates, see Fig. 2. The lowest barriers between equivalent minima are only 5 and 22 cm⁻¹. Such a PES results in an exceptionally fluxional dimer and also explains why finding the minimum of the PES was much more difficult than anybody could have expected in the 1980s and 1990s. Only in the early 2000s theoretical calculations located the minimum and the lowest stationary points with a reasonable accuracy.

We have achieved excellent agreement of our purely first-principles predictions with the most direct outcomes of spectral experiments such as VRT transition frequencies (RMSD of about 1 cm⁻¹) and rotational constants (0.1%), so our computational model describes the ammonia dimer rather faithfully. Also observables extracted from the measurements, such as the average orientations of the monomers in the ground rovibrational level and the dipole moment, are in excellent agreement with experiment: the angular parameters agree to within 1.5° and the dipole moment to 0.05 Debye. These findings agree well with those of ref. 19, but are more convincing since we have now achieved agreement with experiment using a first-principles PES rather than an empirical one, and with a much higher numerical precision of the spectral calculations.

The high accuracy of our model allows us to reinterpret results which were deduced from measurements. The small differences in the measured properties for different $NH_3$–$NH_3$ isotopologues, with ¹⁴N replaced by ¹⁵N and H by D, were interpreted as indicating a nearly rigid dimer, which led to the presumption that the experimental rovibrationally averaged structure should be close to the equilibrium structure. We show that the smallness of the isotopic differences is actually due to two competing and partly canceling quantum dynamical effects. In fact, both the $NH_3$–$NH_3$ and $ND_3$–$ND_3$ dimers are exceptionally flexible due to flat regions of the PES in the vicinity of the minima. The experimental structure is an average over the equilibrium and low-lying saddle point structures, which is also affected by dynamical effects that depend on the nature of the monomers: *ortho*$NH_3$ or *para*$NH_3$. While only the path through SP#1 has

been considered in the literature, we show that also tunneling through SP#2 is important and that in fact the experimental structure (and our nearly identical theoretically averaged structure) is similar to that of SP#2.

Our work gives the definitive answer to the question whether ammonia hydrogen bonds. Due to the extreme floppiness of the dimer, the answer cannot be just based on the geometry of the global minimum: one has to consider the probability of finding the monomers at geometries that can be considered as hydrogen bonded. Figure 5 shows that, despite the significant probability for the dimer to be at some stationary-point configurations, the largest probability corresponds to the regions of the minima, so we may say that ammonia does hydrogen bond.

## Methods

**Generation of PES.** Our ab initio PES was calculated using CCSD(T) and assumed a rigid-monomer approximation. We used the vibrationally averaged monomer geometry, $\langle r \rangle_0$, from ref. 37, since it has been shown that rigid-monomer PESs obtained with vibrationally averaged geometries of the monomers predict more accurate observables than those using the equilibrium geometries[38,39]. The CCSD(T) interaction energies were first computed using the frozen-core (FC) approximation and extrapolated to the complete basis set (CBS) limit from augmented basis sets of quadruple and quintuple zeta quality. Then the all-electrons (AE) correction for the FC approximation was computed using augmented triple-zeta quality bases. All interaction energies were counterpoise (CP) corrected. This level of electronic structure theory is substantially higher than used in any published work that developed ammonia-dimer potentials. By analyzing the convergence pattern of the interaction energy in basis set size, we estimated the uncertainty of our interaction energies around the minimum of the PES to be 0.015 kcal/mol with respect to the exact value at the CCSD(T) level of theory. We have also computed the contributions of full triple and noniterated quadruple excitations included in CCSDT(Q) using a partly augmented double-zeta quality basis set. We found that the T(Q) contribution is −0.011 kcal/mol. Various other contributions beyond CCSD(T) were evaluated by Boese[40] and are still smaller in magnitude. Thus, we can estimate the uncertainty of our ab initio interaction energies relative to the exact solutions of Schrödinger's equation to be 0.02 kcal/mol in the region of the van der Waals minimum. Further details of the ab initio calculations are given in Supplementary Methods.

Generations of grid points (dimer geometries for which ab initio calculations were performed) and fitting them was done applying the autoPES codes[41], an automatic PES development software. An important feature of this program is the use—as a part of the fit—of an accurate large-$R$ asymptotic expansion computed ab initio from monomer properties. The close-range part of the fit is a sum of isotropic site-site functions. The set of sites included all atoms plus 6 symmetry-unique off-atomic sites with optimized positions. The positions of charged off-atomic sites were optimized by fitting the approximate multipole moments produced by partial charges to ab initio computed COM multipole moments of the monomer, following the approach of ref. 42. The fit was performed to a training data set consisting of 1576 grid points and the root-mean square error (RMSE) of the fit for $E_{\text{int}} < 0$ was 0.0082 kcal/mol. Accounting for the fit RMSE, the uncertainty of the fit relative to the exact interaction energy is still 0.02 kcal/mol (adding uncertainties in squares). The detailed form of the fit function is given in Supplementary Methods. A FORTRAN program computing the fitted potential is included in the Supplementary Data in the file Supplementary_Data_1.zip. This file also contains coordinates and energies of all computed data points, parameters of the fit, and instructions on using the program.

**SAPT interaction energy components.** SAPT calculations for selected regions of the PES were made with the variant of SAPT based on a density-functional description of monomers, SAPT(DFT)[43,44], using the SAPT2020 codes[45] interfaced with ORCA[46], the PBE0 functional[47,48], and an augmented triple-zeta basis set. The SAPT contributions are denoted as $E_{\text{type}}^{(i)}$ where $i$ indicates the order of perturbation theory in the intermolecular interaction operator. The SAPT components discussed in the main text are as follows. Electrostatics: $E_{\text{elst}}^{(1)}$, exchange-repulsion: $E_{\text{exch}}^{(1)}$, induction: $E_{\text{ind}}^{(2)} + E_{\text{exch}-\text{ind}}^{(2)} + \delta E_{\text{int,resp}}^{\text{HF}}$, where the last term collects induction and exchange-induction contributions of order higher than the second computed as the difference between the supermolecular Hartree-Fock (HF) interaction energies and the sum of appropriate SAPT corrections, dispersion: $E_{\text{disp}}^{(2)} + E_{\text{exch}-\text{disp}}^{(2)}$. The coupled Kohn–Sham and coupled HF (response) approaches were used throughout.

**VRT states.** In calculations of VRT states, just as in earlier work on the ammonia, water, and benzene dimers[18,49,50], we determined the lower eigenvalues of the dimer Hamiltonian[51] in a coupled product basis of symmetric rotor functions—

Wigner $D$-functions[52]—for the angular coordinates, multiplied with a set of radial sine-type basis functions. The total angular momentum $J$ is a good quantum number and is held fixed. Its projection $K$ on the dimer axis $\mathbf{R}$ is a nearly good quantum number and can be used in combination with $J$ to label the dimer VRT states (although there is some mixing between basis functions with different values of $K$ by off-diagonal terms in the Coriolis coupling operator). The eigenstates of the $NH_3$–$NH_3$ (or $ND_3$–$ND_3$) rigid-monomer Hamiltonian are calculated with a pseudo-spectral method, implemented by Leforestier et al.[53,54]. In this method, the kinetic energy operators in the Hamiltonian act on the analytic basis, which yields rather simple expressions. The potential matrix is calculated in a six-dimensional grid basis in which it is diagonal; its diagonal elements are simply the values of the potential at the grid points. A set of low-lying eigenvalues and the corresponding eigenfunctions are computed with the Lanczos algorithm[55] that iteratively converges a set of trial vectors. It is not necessary to compute the Hamiltonian matrix; only the action of the terms in the Hamiltonian on the Lanczos vectors is required in each iteration step. The program includes very efficient transformations of the Lanczos vectors from the grid basis to the analytic basis, and vice-versa. Details on the radial and angular basis functions are given in Supplementary Methods.

## Data availability

The data that support the findings of this study are included within the Article and Supplementary Information. In particular, the .zip file contains coordinates and energies of all computed data points and parameters of the fit.

## Code availability

The codes used used for electronic structure calculations and fitting, MOLPRO, MRCC, SAPT, and autoPES (part of SAPT package), are available on the web and the links are provided in references of the main paper and the Supplementary Information. The program computing VRT spectra is available from the authors upon reasonable request. A FORTRAN program computing the fitted potential is included in the Supplementary Data in the file Supplementary_Data_1.zip.

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

## Acknowledgements

This research was supported by NSF grant CHE-1900551 (K.S.). We thank Claude Leforestier for making his computer program available to us.

## Author contributions

A.J. and K.S. performed the ab initio calculations and constructed the $NH_3$–$NH_3$ potential. A.v.d.A. calculated the VRT states and properties of $NH_3$–$NH_3$ and $ND_3$–$ND_3$. All authors were involved in discussing these results and writing the paper, including the artwork for the figures.

## Competing interests

The authors declare no competing interests.
