## [Peer Review File · Nature Communications]

REVIEWER COMMENTS

Reviewer #1 (Remarks to the Author):

The authors present results from careful calculations on the Ammonia dimer, and its deuterated analogue, showing very good agreement with (old) experimental results. The authors are looking for controversy in regards to interpretation of the experiment. I think they are overreaching in doing so.

Firstly, I think the presentation can be clarified somewhat for the reader.

I think the authors should include (a modified) Figure S1 in the main body of the text, including the relative energies. It would also be nice to show some MEP relating the TS to the minimum energy structures (labeling the equivalent atoms for clarity) (perhaps a movie, or some frames for geometries). I think the authors should also include the average structure obtained from calculations. The reader can see how close the averaged geometries are to experiment. Adding the C3 axes on acceptor might help visualization.

Now if the authors would interpret their findings from scratch, not looking for controversy, how would one explain it? I think something along these lines.

I would say the floppiness is extraordinary. Nitrogen atoms move from donor to acceptor and everything in between in a range of 5 cm^{-1} . In another motion the H-atom in the H-bond rotates in and out in a range of 22 cm^{-1} (I think). The nuclear wfn's have significant weight at the transition state regions. All of this is very different from traditional directional Hydrogen bonds.

It is remarkable that there is very little change between the NH_3 and ND_3 dimers. This 'coincidence' led the original experimentalists to their interpretation that the system is rigid. I think any calculation (or reflection) would have pointed to a fluxional system, and therefore a coincidence. More data on different isomers, NH_2D perhaps might confirm i.e. leading to different averaged structures?).

The authors emphasize completely different aspects. They point to the directional H-bonding in the (meaningless ?) equilibrium structures, and downplay in their interpretation (I think) the fact that TS is only 5 cm^{-1} away.

I think the title is hard to justify. This is a remarkable system.

I am not an expert in VRT spectroscopy. I am quite curious how experimentalists deduce structures, rotational constants from their spectroscopy. Usually this involves fitting energy levels to parameters in a model. The authors might say something about this. The level of agreement between theory and experiment is excellent for all deduced quantities (even using erroneous models?). The deduction of interpretation from exp. was quite a bit off and this is fairly easily corrected by theory. I think it shows one has to divorce the interpretation aspects from the hard data. We like to get to simple one-line interpretations. Here: Ammonia does hydrogen bond!! Really? That is the takeaway?

I am not sure what this does for a publication in Nature Chemistry. No controversy, no interest, no publication? I get the impression that the desire to publish in Nature Chemistry distorts the point of view towards whipped-up controversy. This is bad for science in my view. Many papers may suffer from this attention grabbing syndrome. It feels like I am battling windmills here.

.

Reviewer #2 (Remarks to the Author):

Jing et al. present a new highly accurate potential energy surface (PES) for the ammonia dimer, combined with a calculation of the vibrational-rotational-tunneling (VRT) spectrum and an analysis of the energy contributions using symmetry adapted perturbation theory (SAPT). The PES is the most accurate to date and the agreement of the calculated VRT spectrum with experiment is excellent. The authors demonstrate the importance of two very low-lying saddle points on the PES for the experimental geometry of the ammonia dimer resulting from quantum mechanical averaging over the wave function which extends substantially towards the two saddle points resulting in a substantial difference of the PES equilibrium structure and the experimentally observed structure.

The paper represents a substantial contribution toward the understanding of the ammonia dimer as the known difference between experimental and equilibrium structure is now well analyzed and thus understood. This work is of substantial significance as the question of the structure of the ammonia dimer and whether it is hydrogen bonded is a long standing fundamental question in spectroscopy and theoretical chemistry. The authors combine several state-of-the-art methods expertly and describe their work in great detail in the supporting information.

The paper should be accepted after a minor revision that addresses the following issues.

(1) I find the presentation of the stationary points very unfortunate making it difficult for the reader to follow. The two paths connecting the PES minimum and the two lowest saddle points can be described and visualized easily as donor-acceptor exchange and monomer rotation about the C3 axis. This is mentioned very briefly in the manuscript but not visualized. I would recommend a figure with the PES minimum and arrows indicating the two motions, or alternatively placing the Fig S1 into the main text.

(2) p.5 "The significant variation of the exchange-repulsion energy along the path also explains the narrowness of the valley".

I cannot follow that argument. The narrowness refers to the PES gradient orthogonal to the path, so I cannot see why it can be explained with contributions along the path. It is important for the context to understand this narrowness.

(2) p.5 "is almost opposite of that discussed above for the linear structure". I cannot find the mentioned discussion.

Reviewer #3 (Remarks to the Author):

The value of the current study lies the construction of a state-of-the-art potential energy surface (PES) for the ammonia dimer and the subsequent computation of the measured Vibration-Rotation-Tunneling (VRT) spectra, producing excellent agreement with the experimental values. However, it has not changed, but rather confirmed, the prevailing view of the community regarding the structure of the ammonia dimer, namely that it is hydrogen bonded. As stated in the abstract, "We confirm earlier calculations showing that the equilibrium structure of the dimer is hydrogen bonded." In the conclusions section, "(our) conclusions generally agree with those in Ref. 19, but are more convincing ..." and "(the current study) has now achieved agreement with experiment in a first-

principles approach and with a much higher numerical precision of the spectral calculations.” Even the original authors of the 1987 Science paper (reference 9), which reported an inaccurate structure for the ammonia dimer that the current study as well as numerous previous ones have succeeded in contradicting, have questioned their own findings in a subsequent publication in 1993 (Reference 24) by stating: “... in all likelihood rules out the original suggestion made by one of us that this was the equilibrium form.”

In my opinion the paper is more suited for a more specialized journal in the field of Chemical Physics / Physical Chemistry as it confirms previous work on the system using a much more accurate PES. The title of the manuscript “Ammonia does hydrogen bond!” (why the exclamation mark, is that a surprise?) is stating what is generally accepted in the community about this system.

Comments of major significance:

1) The PES developed in this study is based on rigid monomers (see p. 3) resulting in an overall (but still impressive) 6-dimensional VRT calculation. Given the excellent agreement between the computed and measured VRT spectra, I wonder what the effect of inter-molecular fragment geometry relaxation on the computed VRT spectral values is, especially for regions where a hydrogen atom is in the area between the two nitrogen atoms. This will also affect some barriers reported in the study. Perhaps this effect is minimal, however, it would help if the authors can provide an estimate of its magnitude.

2) Besides the VRT spectra, which can be directly attributed to the underlying structure, another important “hallmark” of the presence (or not) of a hydrogen bond is the overall structure (peaks) and position of the infrared active fundamental bands. The current study does not address this important property at all.

3) The full form of the used PES is not available. The functional form is given in Section II of the SI but not the values of the fitted parameters! This is important for other researchers to reproduce the results reported. The authors should provide a subroutine that returns the value of V , equation (S1) in the SI, from the Cartesian coordinates of all atoms as input.

Additional comments of minor significance:

1) The importance of the ammonia dimer on p. 2 (text before Figure 1) is exaggerated. It is the ammonia monomer that is important in terrestrial and extraterrestrial environments as well as in industrial and pharmaceutical synthesis, in living organisms and as a green fuel, NOT its dimer!

2) There seems to be missing text in the beginning of the last paragraph of p. 7: the paragraph starts “states and the second G state.”

Recommendation: publish in a more specialized journal.

Report of Reviewer 1

The authors present results from careful calculations on the Ammonia dimer, and its deuterated analogue, showing very good agreement with (old) experimental results. The authors are looking for controversy in regards to interpretation of the experiment. I think they are overreaching in doing so.

The disagreement between theory and experiment that started in the 1980s is a fact and it would be hard to deny it. So perhaps the word “controversy”, used by us only a few times in the manuscript, sounds too strong for the reviewer. To avoid this issue, we changed all these occurrences into “disagreement”. On a more fundamental level, we believe that the opinions of Reviewer 1 are actually very close to ours. Most of our manuscript is devoted to the discussion of the unusual properties of the ammonia dimer. Only a fraction of it discusses the controversies.

Firstly, I think the presentation can be clarified somewhat for the reader.

I think the authors should include (a modified) Figure S1 in the main body of the text, including the relative energies. It would also be nice to show some MEP relating the TS to the minimum energy structures (labeling the equivalent atoms for clarity) (perhaps a movie, or some frames for geometries). I think the authors should also include the average structure obtained from calculations. The reader can see how close the averaged geometries are to experiment. Adding the C3 axes on acceptor might help visualization.

We have moved Fig. S1 to the main text and modified it as requested. In particular, we added to it the averaged theoretical structure, included the C₃ axes on all dimers, and added arrows showing the directions of motion from the two stationary-point dimers leading to the minimum dimer. We hope the latter is sufficient to visualize the minimum energy path between these structures. For the path from SP#2 to SP#0, we have additionally included in the SI a series of frames showing this motion (see the new Fig. S1).

Now if the authors would interpret their findings from scratch, not looking for controversy, how would one explain it? I think something along these lines.

I would say the floppiness is extraordinary. Nitrogen atoms move from donor to acceptor and everything in between in a range of 5 cm⁻¹. In another motion the H-atom in the H-bond rotates in and out in a range of 22 cm⁻¹ (I think). The nuclear wfn's have significant weight at the transition state regions. All of this is very different from traditional directional Hydrogen bonds.

It is remarkable that there is very little change between the NH₃ and ND₃ dimers. This ‘coincidence’ led the original experimentalists to their interpretation that the system is rigid.

We agree with the reviewer on all these points and we believe the manuscript has already stated them. We have made a few revisions to emphasize these points even more. In particular, on p. 3 we related the flatness of the PES in the region of the minima to the floppiness of the dimer and extended the discussion of floppiness in the Conclusions.

I think any calculation (or reflection) would have pointed to a fluxional system, and therefore a coincidence.

We believe that no basis for such a conclusion existed until the present work. Calculations of the PES capable to determine that it is so flat around the minimum were not possible in the 1980s and 1990s and in fact our work presents the first such PES. While one may find speculations in the literature that the ammonia dimer is highly fluxional, only our calculations provide convincing evidence. For example, the results from a limited sampling performed in Ref. 24 indicated that the minimum region of the PES is flat, but the global minimum was found incorrectly at a C_{2h} configuration (which is not hydrogen bonded), so also the findings concerning flatness could be questioned. A similar text, except for the last sentence, was added to the Conclusions.

More data on different isomers, NH₂D perhaps might confirm i.e. leading to different averaged structures?

We could perform such calculations, but we believe the evidence and the explanation of the isotope effects is already very convincing. The change from NH_3-NH_3 to ND_3-ND_3 represents the most substantial isotopic effect, significantly larger than the effect of replacing NH_3 by NH_2D . Moreover, the NH_2D dimer has a different PI group, so comparisons between NH_3 , ND_3 , and NH_2D isotopologues would not be straightforward.

The authors emphasize completely different aspects. They point to the directional H-bonding in the (meaningless ?) equilibrium structures, and downplay in their interpretation (I think) the fact that TS is only 5 cm⁻¹ away.

We believe we do emphasize the very small, 5 and 22 cm^{-1} , barriers between minima. We added an explicit statement about it in the Conclusions. However, we think the equilibrium structure is not meaningless. As Fig. 5 shows, the probability of the dimer to be at the equilibrium geometry is still the prevailing probability, higher than at the “experimental” (or theoretically averaged) geometry. Tunneling between symmetry-equivalent equilibria is present in essentially all hydrogen-bonded dimers of small molecules, although the tunneling splittings are much larger here. A similar text was added to the Conclusions.

I think the title is hard to justify. This is a remarkable system.

We have changed the title into: “Ammonia dimer: extremely fluxional but still hydrogen bonded”.

I am not an expert in VRT spectroscopy. I am quite curious how experimentalists deduce structures, rotational constants from their spectroscopy. Usually this involves fitting energy levels to parameters in a model. The authors might say something about this. The level of agreement between theory and experiment is excellent for all deduced quantities (even using erroneous models?). The deduction of interpretation from exp. was quite a bit off and this is fairly easily corrected by theory. I think it shows one has to divorce the interpretation aspects from the hard data.

Extraction of the molecular geometry from measured spectra is usually done by extracting the rotational constants of the molecule from rotationally resolved high-resolution spectra. The knowledge of only three rotational constants is generally not sufficient to determine the positions of all atoms in the molecule, and one also measures the spectra and extracts the rotational constants of various isotopologues. With the assumption that the structure of a semi-rigid molecule is hardly changed by isotope substitution, one can then determine the atomic positions in the molecule. In their measurements on the ammonia dimer, NFK

found that the dipole moment and the nuclear quadrupole splittings (which determine the orientations of the monomer C_3 axes in the dimer) hardly change upon isotope substitution. Hence, they concluded that the dimer is nearly rigid, so that the structure deduced from these experimental data must be close to the dimer equilibrium geometry. We added a comment to this effect in the Conclusions. As an aside, we agree of course that the interpretation of measured results can be wrong, but without interpretations progress in science would be almost impossible.

We like to get to simple one-line interpretations. Here: Ammonia does hydrogen bond!! Really? That is the takeaway?

First, let us point out that Reviewer 3 believes that ammonia is obviously hydrogen bonded. A simple one-line interpretation is not possible in this case. However, our point is that although the ammonia dimer is highly fluxional, it still can be considered to be hydrogen bonded based on the probability of finding it in the hydrogen-bonded structure, as argued above. This conclusion is different from that of NFK.

I am not sure what this does for a publication in Nature Chemistry. No controversy, no interest, no publication? I get the impression that the desire to publish in Nature Chemistry distorts the point of view towards whipped-up controversy. This is bad for science in my view. Many papers may suffer from this attention grabbing syndrome. It feels like I am battling windmills here.

We do believe there was a controversy (although we now call it discrepancy) and that our work resolves it. In the revision, we have tried to emphasize other important findings of our work, such as the exceptionally fluxional character of the dimer and the surprising cancellation of nuclear motion quantum effects. Incidentally, we also thought that our manuscript is best suited for Nature Chemistry, but the Nature editors suggested its transfer to Nature Communications.

Report of Reviewer 2

Jing et al. present a new highly accurate potential energy surface (PES) for the ammonia dimer, combined with a calculation of the vibrational-rotational-tunneling (VRT) spectrum and an analysis of the energy contributions using symmetry adapted perturbation theory (SAPT). The PES is the most accurate to date and the agreement of the calculated VRT spectrum with experiment is excellent. The authors demonstrate the importance of two very low-lying saddle points on the PES for the experimental geometry of the ammonia dimer resulting from quantum mechanical averaging over the wave function which extends substantially towards the two saddle points resulting in a substantial difference of the PES equilibrium structure and the experimentally observed structure.

The paper represents a substantial contribution toward the understanding of the ammonia dimer as the known difference between experimental and equilibrium structure is now well analyzed and thus understood. This work is of substantial significance as the question of the structure of the ammonia dimer and whether it is hydrogen bonded is a long standing fundamental question in spectroscopy and theoretical chemistry. The authors combine several state-of-the-art methods expertly and describe their work in great detail in the supporting information. The paper should be accepted after a minor revision that addresses the following issues.

We thank the reviewer for such an accurate characterisation of our work.

(1) I find the presentation of the stationary points very unfortunate making it difficult for the reader to follow. The two paths connecting the PES minimum and the two lowest saddle points can be described and visualized easily as donor-acceptor exchange and monomer rotation about the C3 axis. This is mentioned very briefly in the manuscript but not visualized. I would recommend a figure with the PES minimum and arrows indicating the two motions, or alternatively placing the Fig S1 into the main text.

We have exactly followed the reviewer’s directions, see the answer to Reviewer 1.

(2) p.5 “The significant variation of the exchange-repulsion energy along the path also explains the narrowness of the valley”. I cannot follow that argument. The narrowness refers to the PES gradient orthogonal to the path, so I cannot see why it can be explained with contributions along the path. It is important for the context to understand this narrowness.

We have rewritten the questioned sentence and added a graph in the SI, Fig. S3, showing the changes of the interaction energy components when the system moves in the direction orthogonal to the valley in Fig. 2(a).

(2) p.5 “is almost opposite of that discussed above for the linear structure”. I cannot find the mentioned discussion.

This was poor grammar; the sentence was rewritten to make it clear. What we meant was the difference between the linear structure and SP#1, both shown in Fig. 3.

Report of Reviewer 3

The value of the current study lies in the construction of a state-of-the-art potential energy surface (PES) for the ammonia dimer and the subsequent computation of the measured Vibration-Rotation-Tunneling (VRT) spectra, producing excellent agreement with the experimental values. However, it has not changed, but rather confirmed, the prevailing view of the community regarding the structure of the ammonia dimer, namely that it is hydrogen bonded.

We disagree concerning the “prevailing view”. Also Reviewers 1 and 2 disagree. Also, we have shown that the ammonia dimer is extremely fluxional and that the hydrogen bond is not the usual one. This is further explained and emphasized in the revised version.

As stated in the abstract, “We confirm earlier calculations showing that the equilibrium structure of the dimer is hydrogen bonded.”

Here we meant only the position of the minimum on the PES. It was still possible that the probability of finding the dimer in such a configuration is low. This statement has now been made more explicit.

In the conclusions section, “(our) conclusions generally agree with those in Ref. 19, but are more convincing” and “(the current study) ha(s) now achieved agreement with experiment in a first-principles approach and with a much higher numerical precision of the spectral calculations.”

Perhaps we tried to give too much credit to the previous work. We added a few sentences in the Conclusions pointing out substantial differences with previous work.

Even the original authors of the 1987 Science paper (reference 9), which reported an inaccurate structure for the ammonia dimer that the current study as well as numerous previous ones have succeeded in contradicting, have questioned their own findings in a subsequent publication in 1993 (Reference 24) by stating: “in all likelihood rules out the original suggestion made by one of us that this was the equilibrium form.”

We cannot agree that “numerous previous [studies] have succeeded in contradicting” the findings of NFK. There were indeed many electronic structure calculations (some cited in our manuscript) that have been improving the energy and geometry of the equilibrium structure. These studies were only making the discrepancies with experiment more severe. The only previous study that addressed the discrepancies was a series of papers coauthored by one of us. However, the main conclusions were based on calculations using an empirical PES fitted to experimental spectra, which limited the reliability of the conclusions. The discussion of this issue in the manuscript has been extended.

One can formulate the same point in a different way. The most recent electronic structure calculations determined the dimer equilibrium geometry to high precision. This does not mean that they contradicted experiment. None of these calculations determined the probability for the dimer to be found at the equilibrium structure relative to the experimental one. Had the latter been larger than the former, the ammonia dimer could not be considered to be hydrogen bonded.

There are other physical quantities that electronic structure calculations cannot fully predict, for example the dimer dipole moments. While computed and experimental dipole moments are in good agreement for most dimers, e.g., for the water dimer, for the ammonia dimer the directly measured dipole moment is 0.74 D, while the dipole moment at equilibrium from our calculations is 1.64 D, both results accurate to within about 0.05 D. Clearly, there is no way electronic structure calculations can resolve this discrepancy.

Concerning Ref. 24: yes, as already discussed above, the results of Ref. 24 indicated (based on a very limited sampling) that the PES is flat in the region of the minimum, but the problem with this work is that the equilibrium structure found is incorrect.

In my opinion the paper is more suited for a more specialized journal in the field of Chemical Physics / Physical Chemistry as it confirms previous work on the system using a much more accurate PES. The title of the manuscript “Ammonia does hydrogen bond!” (why the exclamation mark, is that a surprise?) is stating what is generally accepted in the community about this system.

As already explained, some previous work could not really be trusted since its main conclusions were based on an empirical PES fitted to the spectra, while previous electronic structure calculations do not really address the important issues. We also note that the Reviewers 1 and 2 would probably not support the opinion on ammonia hydrogen bonding to be a generally accepted paradigm.

Comments of major significance:

(1) The PES developed in this study is based on rigid monomers (see p. 3) resulting in an overall (but still impressive) 6-dimensional VRT calculation. Given the excellent agreement between the computed and measured VRT spectra, I wonder what the effect of inter-molecular fragment geometry relaxation on the computed VRT spectral values is, especially for regions

where a hydrogen atom is in the area between the two nitrogen atoms. This will also affect some barriers reported in the study. Perhaps this effect is minimal, however, it would help if the authors can provide an estimate of its magnitude.

Nowadays one could compute an accurate full-dimensional potential surface for the ammonia dimer. Nuclear dynamics calculations that include the monomer-flexibility effects are still impossible, however, as the full-dimensional potential has 18 degrees of freedom. For the water dimer with 12 degrees of freedom such investigations have recently become possible, and Fig. 2 in Ref. 36 compares the accuracy of 6D and 12D calculations for this system. This figure shows that there is no systematic improvement of the intermolecular VRT levels when going from 6D to 12D. For the ammonia dimer, one option could be to include only the umbrella motion in the monomers (model calculations were presented in Ref. 20). This would be an important, but difficult research project. We have extended the discussion of monomer flexibility effects (now on p. 7-8).

(2) Besides the VRT spectra, which can be directly attributed to the underlying structure, another important “hallmark” of the presence (or not) of a hydrogen bond is the overall structure (peaks) and position of the infrared active fundamental bands. The current study does not address this important property at all.

We assume the reviewer means the red or blue shifts of the monomer infrared transitions caused by intermolecular interactions in the dimer. We have not discussed these properties since an all-dimensional PES would be needed for such calculations. We actually cited two experimental papers in this field, Refs. 12 and 14. We do not believe that these results shed light on the question of hydrogen bonding in the ammonia dimer. If it were not hydrogen bonded, the strength of the bond would remain nearly the same, and this strength determines the shifts.

(3) The full form of the used PES is not available. The functional form is given in Section II of the SI but not the values of the fitted parameters! This is important for other researchers to reproduce the results reported. The authors should provide a subroutine that returns the value of V , equation (S1) in the SI, from the Cartesian coordinates of all atoms as input.

The requested data are now included in the SI.

Additional comments of minor significance:

(1) The importance of the ammonia dimer on p. 2 (text before Figure 1) is exaggerated. It is the ammonia monomer that is important in terrestrial and extraterrestrial environments as well as in industrial and pharmaceutical synthesis, in living organisms and as a green fuel, NOT its dimer!

Indeed we meant the importance of the ammonia molecule. This fragment has been revised.

(2) There seems to be missing text in the beginning of the last paragraph of p. 7: the paragraph starts “states and the second G state.”

This text was not missing, it was just above Fig. 4 (the paragraph is separated by Fig. 4).

Recommendation: publish in a more specialized journal.

We believe that this work and its conclusions are of sufficient general importance for a broad readership journal.

REVIEWERS' COMMENTS

Reviewer #1 (Remarks to the Author):

I am fully satisfied by the changes made by the authors. We were probably closer in regards to content as I thought in my first review. I think this is a very nice paper, suitable for publication in Nature Communications.

Reviewer #3 (Remarks to the Author):

In the revised manuscript the authors have addressed most of the serious comments suggested by the reviewers and have toned down the language in the title and manuscript regarding the significance of the paper. Most importantly, they have added a subroutine that returns the energy of the fitted potential given the Cartesian coordinates. The issue of also reproducing the IR spectra still remains but the paper can be accepted without these at this stage.

Report of Reviewer 1 *I am fully satisfied by the changes made by the authors. We were probably closer in regards to content as I thought in my first review. I think this is a very nice paper, suitable for publication in Nature Communications.*

Report of Reviewer 3 *In the revised manuscript the authors have addressed most of the serious comments suggested by the reviewers and have toned down the language in the title and manuscript regarding the significance of the paper. Most importantly, they have added a subroutine that returns the energy of the fitted potential given the Cartesian coordinates. The issue of also reproducing the IR spectra still remains but the paper can be accepted without these at this stage.*

We are happy that all reviewers now advise to publish our paper in Nature Communications in its present form. The FORTRAN code of the fitted potential has been included in the Supplementary Information.